

# Hematological parameters in largemouth bass (*Micropterus salmoides*) with formalin-preservation: comparison between wild tournament-caught and captive-raised fish

Michelle C. Whitehead[1], Chelsey L. Vanetten[1], Yaxin Zheng[2] and Gregory A. Lewbart[1]

[1] Department of Clinical Sciences, North Carolina State University, Raleigh, NC, United States of America
[2] Department of Statistics, North Carolina State University, Raleigh, NC, United States of America

Corresponding author
Michelle C. Whitehead,
mcwhite7@ncsu.edu

## ABSTRACT

**Background**. Largemouth bass (*Micropterus salmoides*) are an economically important freshwater fish species that have been investigated for both the short and long-term effects of stress, secondary to angling. Limited data has been published on the hematological parameters of this species and blood sample stability is a notable limitation of hematologic field studies. A relatively novel technique using 10% neutral buffered formalin preserves heparinized whole blood and maintains blood cell stability beyond one month in striped bass. The objective of this study was to evaluate the differences in hematological parameters between tournament-caught and captive-raised largemouth bass using whole blood preservation with neutral buffered formalin.

**Methods**. Two populations of largemouth bass ($n = 26$ wild; $n = 29$ captive) underwent coccygeal venipuncture to collect heparinized whole blood for packed cell volume, total solids, and manual differential. Formalin preservation of heparinized whole blood facilitated manual hemocytometer analysis. Results were compared between the populations (tournament-caught, and captive-raised) with Wilcoxon rank sum test, a Hotelling's $T^2$ test, and Bonferroni simultaneous 95% confidence intervals to determine significance.

**Results**. The mean packed cell volume ($44.9 \pm 5.4\%$) and total solids ($7.2 \pm 1.1$ g/dL) were significantly higher, while the total leukocyte count ($7.08 \pm 1.86 \times 10^3/\mu$L) was significantly lower in the wild tournament-caught population of largemouth bass, as compared to the captive-raised counterparts (PCV $34.4 \pm 7.2\%$; TS $5.2 \pm 1.0$ g/dL; WBC $16.43 \pm 8.37 \times 10^3/\mu$L). The wild population demonstrated a significantly distinct leukogram characterized by a neutropenia ($24.1 \pm 12.7\%$), lymphocytosis ($67.7 \pm 13.0\%$), and monocytopenia ($8.3 \pm 2.9\%$), while the erythrocyte and thrombocyte counts were not significantly different between populations.

**Discussion**. Numerous factors have been demonstrated to influence hematologic parameters in fish including age, size, sex, temperature, environmental oxygen level, population density, and infection. The wild population endured stress during angling capture, live-well hypoxia, transport, and extended air exposures at weigh in, which may have caused a stress leukopenia as well as osmoregulatory dysfunction and subsequent

hemoconcentration. Further evaluation of seasonal impact as well as increased sample size is warranted to enhance our understanding of largemouth bass hematology.

**Conclusion**. This study concluded that wild largemouth bass captured via tournament angling have higher packed cell volume and total solids, and lower total leukocyte counts, compared to captive-reared individuals. Through the completion of this study, we demonstrated the successful use of 10% neutral buffered formalin to preserve heparinized whole blood for precise hemocytometer cell counts in a new teleost species, the largemouth bass.

# INTRODUCTION

Largemouth bass (*Micropterus salmoides*) are an economically important freshwater fish species both for recreational and competitive (tournament) angling. This has given rise to scientific investigations into the impact of angling on a variety of factors affecting the fitness of largemouth bass (LMB), both short and long-term. Studies simulating tournament capture have elicited stress responses, which are related to the degree of exhaustion and water temperature, while often compounded by the length of air exposure, live-well disturbances, and tournament weigh-ins among other factors (*Cooke et al., 2002*). Short-term consequences are associated with risk of immediate death, as well as a negative energetic balance, which adversely impacts reproductive success due to a reduction in locomotory activity and subsequent diminished nest-guarding behavior (*Cooke et al., 2000*). The population level effects of catch-and-release angling of black bass (*Micropterus* spp.) have been summarized for nesting fish, tournament-associated mortality, tournament-related fish displacement, as well as barotrauma, livewell, and weigh-in procedural management (*Siepker et al., 2007*). *Siepker et al. (2007)* discuss the differences between brief catch-and-release events during recreational angling and competitive angling where a fish commonly endures extended stress including capture, air exposure, placement in livewell for variable length of time, transport to distant location for weigh-in measurements, and then transport for release. Based on previous studies, this can prolong the stressful encounter up to 8 h in some instances, while introducing additional air exposure, potential overcrowding and hypoxia in the livewell, as well as displacement at the time and location of release (*Cooke et al., 2002*).

Extrinsic stress (e.g., population density, handling, transport, etc.) can have an acute impact on the body based on physiologic response to the stressor inducing the release of catecholamines followed by corticosteroids (*Clauss, Dove & Arnold, 2008*). If the stressor remains, it is likely to lead to a chronic effect that subsides as acclimation occurs (*Clauss, Dove & Arnold, 2008*). Many variables impact fish hematologic parameters, such as species, age, size, diet, season, and oxygen levels (*McCarthy, Stevenson & Roberts, 1975*; *Clark,*

*Whitmore Jr & McMahon, 1979*; *Hrubec, Smith & Robertson, 2001*; *Subhadra et al., 2006*; *Clauss, Dove & Arnold, 2008*; *Gaulke et al., 2014*). Selected normal LMB hematologic parameters have been reported identifying a positive correlation of hematocrit and hemoglobin with fish length and age, while mean corpuscular hemoglobin concentration was negatively correlated with age (*Clark, Whitmore Jr & McMahon, 1979*). This report however, did not encompass leukocyte or thrombocyte counts (*Clark, Whitmore Jr & McMahon, 1979*). Seasonality has been documented to impact hematocrit and hemoglobin levels of striped bass (*Morone saxatilis*) from the same reservoir, whereby those sampled in the spring had significantly higher values compared to those sampled in the summer (*Tisa, Strange & Peterson, 1983*). Water quality may have a direct impact as well, which is evidenced by LMB maintained in a lower oxygen environment for nearly 2 months, leading to acclimation by means of significantly increasing the hematocrit and hemoglobin concentrations (*Gaulke et al., 2014*). Atlantic sturgeon (*Acipenser oxyrinchus oxyrinchus*) captive raised at three different hatcheries were shown to have varied hematologic and biochemical parameters related to environmental conditions and warranted the use of individual reference ranges per population (*Matsche et al., 2014*).

Fish hematology poses several challenges associated with the presence of nucleated erythrocytes mandating manual cell counts by experienced analysts, and restricted sample viability due to rapid cellular distortion, deterioration, and/or aggregation (*Arnold, Matsche & Rosemary, 2014*). Laboratory access and number of samples compound these challenges, especially in field study conditions. *Arnold, Matsche & Rosemary (2014)* investigated elongation of cell stability in striped bass (*M. saxatilis*) using 10% neutral buffered formalin. The study concluded that preservation of freshly collected heparinized whole blood from striped bass in 10% formalin could reliably maintain the cell morphology for manual hemocytometer counts for upwards of 1 month (*Arnold, Matsche & Rosemary, 2014*).

To the authors' knowledge, complete hematologic parameters in LMB have not been described, nor have the impacts of tournament capture on hematologic parameters been compared to a captive population. The objective of this prospective study was to compare hematologic parameters of tournament-captured and captive-raised LMB assessed via manual hemocytometer using formalin preserved heparinized whole blood. We hypothesized that there would be hematologic parameter disparity between the tournament-captured and captive raised LMB, such that the total leukocyte count would be higher in the tournament-captured individuals, while the packed cell volume and total solids would be higher in those captive-raised.

## MATERIALS & METHODS

### Animals

Two different populations of fish were evaluated in this study. Twenty-six wild LMB were obtained during a catch-and-release tournament hosted by Bass Pro Shops at Falls Lake State Recreation Area, North Carolina (NC) on October 21, 2017. Blood samples were obtained from this wild population in a competitive public setting; just one sample coagulated and was excluded from analysis. Foster Lake and Pond Management, Inc.,

Garner, NC provided 35 captive reared LMB for the study. Not all blood parameters were included for each animal due to insufficient sample, sample coagulation, or sample processing errors. These fish were sourced from an inspected facility that has not had OIE reportable pathogens detected via regulatory or clinical sample testing (Farm Cat, Inc., Lonoke, AR, USA). The Foster Lake and Pond Management, Inc. facility was composed of numerous 1,000-gallon (3,780 L) tanks each containing 800 gallons (3,024 L) of water and approximately 500 fish. A multifaceted filtration system including a bubble bead filter and ultraviolet light was utilized in addition to three back-flushes per week with a commercial sand filter. The water temperature was maintained at approximately 17 degrees Celsius, LED lighting was provided 8 h per day, and the diet consisted of maintenance Purina Aquamax 600. This second population of fish were captured and sampled in a controlled setting on April 7, 2018, with minimization of potential stressors. The study protocol was approved by the Institutional Animal Care and Use Committee at North Carolina State University College of Veterinary Medicine (Protocol #18-009-O).

## Sample collection

Based on outward observation, body condition, and gross gill appearance, the fish included in the study were robust without any external lesions. Venipuncture was performed via manual restraint from the lateral approach to the caudal vein. Heparinized sterile 3-mL luer-lock syringes with 22–25-gauge needles were used, depending on the fish size, to collect approximately 0.5–1.0 mL blood. Blood smears and hematocrit tubes were immediately prepared in duplicate.

## Sample processing

The hematocrit tubes were centrifuged at 13,300 RPM for 2 min, followed by measurement of packed cell volume and total solids via refractometer within 8 h of collection. The slides were fixed and stained with Wright-Giemsa stain within 48 h of slide preparation using the North Carolina State College of Veterinary Medicine (NC State-CVM) Clinical Pathology laboratory. The heparinized whole blood was mixed via manual tube inversion and remained at ambient temperature (approximately 22 degrees Celsius) until all samples had been collected. Within 1 h of sample collection, preservation with formalin was completed in duplicate at each collection site, as defined by *Arnold, Matsche & Rosemary (2014)*, whereby 100 μL aliquot of well-mixed heparinized whole blood was placed into 400 μL of 10% neutral buffered formalin for a 1:5 dilution and thoroughly mixed via multiple manual tube inversions. The samples were transported at approximately 22 degrees Celsius within 4 h of collection and stored at 4 degrees Celsius for later sample analysis. Dilution of the blood with Natt-Herrick diluent was adjusted to account for the neutral buffered formalin, such that the final dilution of 1:100 was prepared, per previous publication (*Arnold, Matsche & Rosemary, 2014*). The standard formulas for the Neubauer hemocytometer (Brightline, Hausser Scientific, Horsham, PA, USA) were used as previously documented (*Arnold, Matsche & Rosemary, 2014*). In accordance to *Arnold, Matsche & Rosemary (2014)* (G Lewbart & J Arnold, pers. comm., 2017), the stability of the cells in neutral buffered formalin is minimally time sensitive, exceeding the reported 1 month interval. As such, the
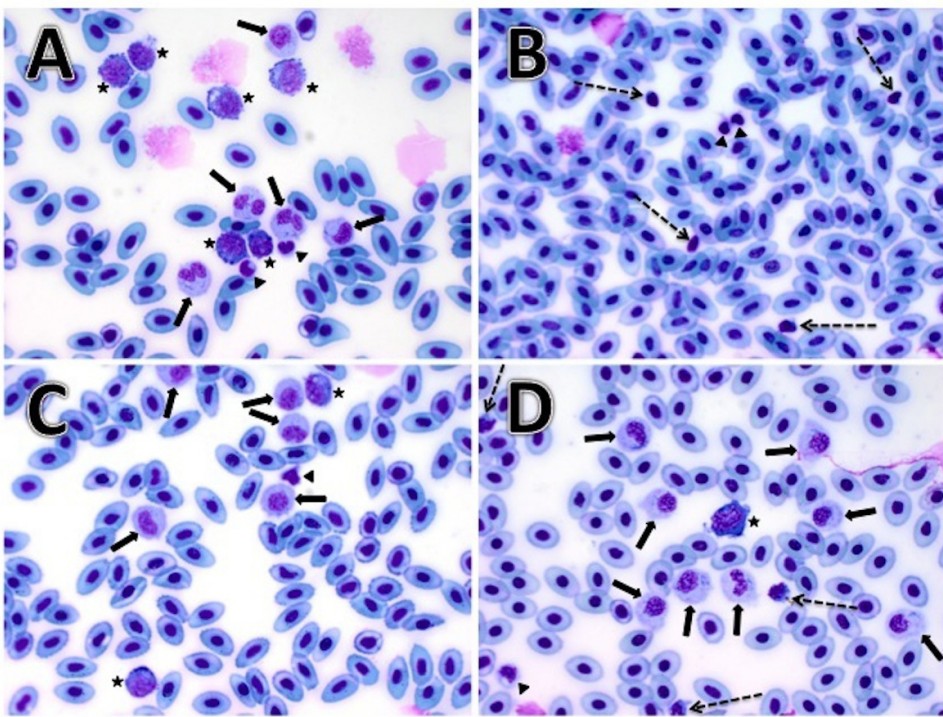

**Figure 1  Microphotograph images demonstrating largemouth bass (*Micropterus salmoides*) hematologic cell morphology with Wright-Giemsa stain at 500× magnification (50× power).** Neutrophils (solid arrows) are identified showing pale cytoplasm with sparse granulation, and an eccentric bi-lobed or unsegmented nucleus. Monocytes (stars) are identified showing the typical eccentric nucleus, abundant vacuolated and deeply basophilic cytoplasm with occasional blebbing of the plasma membrane. Lymphocytes (dashed arrows) are identified with a heterogenous nucleus, scant amount of deeply basophilic cytoplasm and occasional blebbing of the plasma membrane. Thrombocytes (triangle arrowheads) are identified with an irregularly shaped, deeply homogenous basophilic nucleus, and pale cytoplasm, often with cytoplasmic spindle-like extensions. Note that the thrombocytes in these largemouth bass samples did not demonstrate the typical piriform or spindle-form shape, but had more abundant cytoplasm in comparison to the lymphocytes. (A) Representative images of neutrophils (solid arrows), monocytes (stars), and thrombocytes (triangle arrowheads). (B) Lymphocytes (dashed arrows) distinguished from thrombocytes (triangle arrowheads) by the scant and deeply basophilic cytoplasm. (C) Neutrophils (solid arrows), monocytes (stars), and a single thrombocyte (triangle arrowhead). (D) Neutrophils (solid arrows), lymphocytes (dashed arrows), a monocyte (star), and a single thrombocyte (triangle arrowhead).

manual hemocytometer analysis was performed within 3.5 months of sample collection. Additionally, previous investigation of fresh versus formalin preserved blood in striped bass yielded comparable data, and as such, fresh hemocytometer values were not evaluated in the LMB populations of this study (*Arnold, Matsche & Rosemary, 2014*). The manual differential leukocyte counts were obtained via an average of 10 fields at 40× objective multiplied by 1,500, while the manual thrombocyte counts were obtained via an average of 10 fields at 100× objective multiplied by 1,000 as reported (*Ballard & Cheek, 2016*). Microphotographs of representative cells were captured for inclusion in this publication (Fig. 1). The hemocytometer analyses and manual differentials were performed by an experienced licensed veterinary technician (CLV).

**Table 1** Morphometric parameters of mean weight and length (±SD), measured on the day of blood sample collection for the wild and captive populations of largemouth bass (*Micropterus salmoides*).

| Parameter (unit) | Wild | | | Captive | | |
|---|---|---|---|---|---|---|
| | **n** | **Mean ± SD** | **(Min–Max)** | **n** | **Mean ± SD** | **(Min–Max)** |
| **Weight** (kg) | 26 | $0.90 \pm 0.56^{a}$ | (0.2–2.2) | 29 | $0.42 \pm 0.21^{b}$ | (0.23–1.45) |
| **Length** (cm) | 25 | $43.2 \pm 4.6^{a}$ | (37–53) | 29 | $29.5 \pm 2.8^{b}$ | (24–40) |

**Notes.**
Statistical significance set at $P \leq 0.05$ and a Bonferroni correction accounted for multiple comparisons in the data. Those values in the same row with different superscripts are significantly different.

## Statistical analysis

The means ± SD for weight, length, packed cell volume, total solids, as well as counts for total leukocytes, neutrophils, lymphocytes, monocytes, erythrocytes, and thrombocytes were calculated. Wilcoxon rank sum test was used for comparison between the wild and captive LMB morphometric parameters (weight and length), and cell morphology (neutrophils, lymphocytes, monocytes, and thrombocytes). Significance was set at $P \leq 0.05$ and a Bonferroni correction accounted for multiple comparisons in the data. The analysis was performed in JMP Pro 14 (SAS Institute, Cary, NC, USA). Hotelling's $T^2$ test was performed to compare the differences in the remaining blood value measurements (total leukocytes, erythrocytes, packed cell volume, and total solids) between the wild and the captive LMB samples. We then evaluated the Bonferroni simultaneous 95% confidence intervals to identify which specific variables were the most significant. The analysis was performed in SAS 9.4 (SAS Institute, Cary, NC, USA). Assumptions of independence among the blood samples, equal covariance, and normal distributions were made. Data were visually inspected, $t$-test are robust against violations of normality. No severe skewness or outliers were observed.

## RESULTS

The morphometric parameters of weight and length measured for the wild and captive populations of LMB were statistically different (Table 1). The Hotelling's $T^2$ test was conducted on four variables, including packed cell volume, total solids, erythrocyte count, and leukocyte count, demonstrating a significant difference between these two groups (Table 2). Significant differences were detected in the packed cell volume, total solids, as well as counts of total leukocytes, neutrophils, lymphocytes, and monocytes (Table 3). Total leukocyte count was significantly higher in the captive reared population, while packed cell volume and total solids were significantly higher in the wild population, based on the Bonferroni simultaneous confidence intervals evading zero (Table 4). There was no significant difference between the erythrocyte and thrombocyte values in wild and captive populations. Microphotograph images identify LMB neutrophils, lymphocytes, monocytes, and thrombocytes (Fig. 1).
**Table 2** Summary of Hotelling's $T^2$ test results comparing between the wild and captive-raised largemouth bass (*Micropterus salmoides*) populations.

| Hotelling's $T^2$ test | F | df1 ($p$) | df2 ($n$–$p$) | $p$-value |
| --- | --- | --- | --- | --- |
| 94.377 | 22.088 | 4 | 44 | 4.71 E−10 |

**Table 3** Hematologic results from manual differential and hemocytometer analyses of formalin preserved heparinized whole blood for the wild and captive populations of largemouth bass (*Micropterus salmoides*).

| Parameter (units) | Wild | | | Captive | | |
| --- | --- | --- | --- | --- | --- | --- |
| | $n$ | Mean ± SD | (Min–Max) | $n$ | Mean ± SD | (Min–Max) |
| **PCV** (%) | 26 | 44.9 ± 5.4[a] | (31–53) | 28 | 34.4 ± 7.2[b] | (10–48) |
| **TS** (g/dL) | 26 | 7.2 ± 1.1[a] | (5.6–9.6) | 28 | 5.2 ± 1.0[b] | (3–8.7) |
| **RBC** ($\times 10^3$/μL) | 25 | 194.5 ± 37.4 | (109–278) | 25 | 189.2 ± 54.6 | (74–321) |
| **WBC** ($\times 10^3$/μL) | 25 | 7.08 ± 1.86[a] | (4–10.11) | 25 | 16.43 ± 8.37[b] | (5.44–44.56) |
| **Neutrophils** (%) | 26 | 24.1 ± 12.7[a] | (3–53) | 25 | 35.7 ± 15.8[b] | (7–80) |
| **Lymphocytes** (%) | 26 | 67.7 ± 13.0[a] | (38–90) | 25 | 52.8 ± 14.8[b] | (16–80) |
| **Monocytes** (%) | 26 | 8.3 ± 2.9[a] | (1–16) | 25 | 12.2 ± 4.9[b] | (4–22) |
| **Thrombocytes** ($\times 10^3$/μL) | 25 | 15.3 ± 5.8 | (4–30) | 25 | 18.9 ± 9.8 | (8–53) |

Notes.

Statistical significance set at $P \le 0.05$ and a Bonferroni correction accounted for multiple comparisons in the data. Those values in the same row with different superscripts are significantly different. Packed cell volume (PCV), total solids (TS), red blood cell count (RBC), white blood cell count (WBC).

**Table 4** Summary of 95% Bonferroni simultaneous confidence interval results between the wild and captive largemouth bass (*Micropterus salmoides*) populations.

| Parameter | $d$ | 95% Confidence Intervals (CI) | |
| --- | --- | --- | --- |
| | | Lower Bonferroni CI | Upper Bonferroni CI |
| **PCV** | −10.45 | −14.38 | −4.14 |
| **TS** | −1.97 | −2.87 | −0.87 |
| **RBC** | −5.30 | −51.49 | 37.53 |
| **WBC** | 9.35 | 3.32 | 14.77 |

Notes.

$d$ represents the averages of the difference between the wild and captive population for each hematologic parameter. Packed cell volume (PCV), total solids (TS), red blood cell count (RBC), white blood cell count (WBC).

# DISCUSSION

The current study demonstrates the successful implementation of formalin preserved heparinized whole blood to conduct hemocytometer analysis on LMB and compares the hematological parameters between a wild tournament-caught population and captive-raised counterparts. We hypothesized that the wild population captured in the angling tournament would demonstrate a detectable stress leukogram induced secondary to the stress of capture and subsequent events such as live-well hypoxia, transport, and extended air exposure during weigh in. Additionally, we predicted a higher packed cell volume and total solids to be demonstrated in the captive-raised population. However, contrary to our predictions, the wild population demonstrated higher packed cell volumes, higher total solids, and lower leukocyte counts, leading to the rejection of our hypothesis. Manual

differential revealed significantly higher counts of neutrophils and monocytes as well as lower lymphocyte counts in the captive reared individuals.

During spawning season, wild adult striped bass at a fork length between 37 and 70 cm have been documented to have a packed cell volume of 47.9 ± 10.25% (*Westin, 1978*). These striped bass values are most comparable to the LMB wild population in our study (PCV 44.9 ± 5.35%), but 1.39 times higher than those in our captive population of LMB (PCV 34.4 ± 7.16%). Hybrid striped bass at 19 months of age (mean length 26.2 cm, mean weight 0.166 kg) had packed cell volumes of 29–36%, which yields a mean nearest our LMB captive population, yet is much lower than our wild LMB counterparts (*Hrubec, Smith & Robertson, 2001*). The reported leukocyte count of 19-month-old hybrid striped bass ranges from 12.1 to $13.0 \times 10^3/\mu$L, which is lower than the captive (16.43 $\times 10^3/\mu$L) and higher than the wild ($7.08 \times 10^3/\mu$L) populations of LMB in our study (*Hrubec, Smith & Robertson, 2001*). These differences may be related to age and size of the fish sampled, with both of the populations in the current study being larger and heavier than the reported hybrid striped bass (*Hrubec, Smith & Robertson, 2001*). For example, hematologic changes have been associated with age in growing juvenile hybrid striped bass and juvenile and adult rainbow trout (*McCarthy, Stevenson & Roberts, 1975*; *Hrubec, Smith & Robertson, 2001*). The increases in packed cell volume and total solids, and concurrent decrease in leukocyte count described with maturity, may be consistent with the expected and observed development of hematopoietic tissues, such as the spleen and head kidney (*McCarthy, Stevenson & Roberts, 1975*; *O'Neill, 1989*; *Hrubec, Smith & Robertson, 2001*). In the present study, the wild tournament caught fish were of significantly greater mean length (43.2 ± 4.6 cm) and mean weight (0.90 ± 0.56 kg) compared to the captive individuals (length: 29.5 ± 2.8 cm; weight: 0.42 ± 0.21 kg), which suggests an increased level of maturity for the wild population, subsequently yielding higher packed cell volumes and total solids, with lower leukocyte counts. These age-related changes in hematologic parameters are consistent with that reported in Eastern Atlantic loggerhead sea turtles, whereby increasing hematocrit and plasma proteins, as well as decreasing leukocyte count is seen with age (*Casal et al., 2009*). Additionally, stress response with release of endogenous catecholamines and corticosteroids has been reported to manifest as a leukopenia with lymphopenia and relative granulocytosis in fish (*Clauss, Dove & Arnold, 2008*). Thus, the leukopenia relative to the captive individuals could correspond to the advanced handling in the LMB population with tournament capture, transport, and weigh-in; although, an overall lymphopenia and relative granulocytosis was not apparent in this study.

Numerous environmental factors can impact hematologic parameters, such as water quality, temperature, presence of an underlying infection, and population stocking density. The water quality parameters of the lake and tank sources were not within the scope of this study. However, it is known that dissolved oxygen levels can impact hematologic parameters. Largemouth bass acclimate to low levels of dissolved oxygen by enhancement of hemoglobin and hematocrit levels compared to those held at higher dissolved oxygen levels for 50 days (*Gaulke et al., 2014*). The captive population in our study was receiving oxygen supplementation and this may have dampened the drive to enhance hematocrit in the system. It is important to note that although there was discrepancy between

the wild and captive LMB measured hematocrit, the total erythrocyte counts were not significantly different. This suggests an osmoregulatory stress response in the wild LMB population leading to a degree of hemoconcentration, artificially raising the hematocrit. Hematocrit increased in yearling coho salmon in fresh water due to stress, and the investigators concluded that stress induces a bidirectional flux of ions and water across the gill epithelium dependent on the external salinity (*Redding & Schreck, 1983*). Alteration in gill epithelial permeability is evidenced by a five-fold increase in the efflux of sodium and chloride in goldfish secondary to handling stress (*Eddy & Bath, 1979*). Thus, one could speculate that the elevated hematocrit in the wild population of LMB could be secondary to osmoregulatory dysfunction; however, measurement of osmolality was not a component of this study. The thermal tolerance of LMB was quantified whereby an 8 degree Celsius heat shock produced no differences in the three measured physiological indices of stress between two groups of fish from different thermal environments (*Mulhollem, Suski & Wahl, 2015*). However, other differences in these two populations were not elucidated, such as potential behavioral modifications to avoid suboptimal temperature zones, and earlier spawning to ensure optimal developmental temperature for the young (*Mulhollem, Suski & Wahl, 2015*). In goldfish, water temperature impacts various hematologic parameters, such that warm-adapted fish had higher circulating levels of hemoglobin, hematocrit, neutrophil counts, and eosinophil counts (*Houston & Cyr, 1974*; *Dunn, Murad & Houston, 1989*). Previous studies have noted higher hematocrit and hemoglobin concentrations in male goldfish and that experimental infection with *Aeromonas hydrophila* yielded reductions in red blood cell count, hemoglobin and hematocrit in addition to a shift from lymphocyte to polymorphonuclear cell predominance (*Brenden & Huizinga, 1986*). Additionally, goldfish maintained in high stocking densities for at least 6 months demonstrated reflexive leukocytosis and lymphocytosis with corresponding reductions in hematocrit, hemoglobin, and thrombocyte count (*Burton & Murray, 1979*; *Murray & Burton, 1979*). The captive LMB were reared in a tank with known higher stocking density compared to the wild counterparts. Fish raised in close proximity are established to have increased risk of pathogen amplification and transmission of disease, which is contrary to the wide dispersal of wild populations (*Moffitt, Haukenes & Williams, 2004*). Although the thrombocyte count was not significantly different between the two populations, the captive-reared individuals did reflect a significant leukocytosis, lymphocytosis, and lower packed cell volume comparative to the wild LMB. Thus, pathogen exposure and stressors secondary to the captive conditions of close proximity could contribute to the observed lymphocytic leukocytosis, and reduced packed cell volume in the current captive study population.

The precision of fish hematologic cell counts diminishes rapidly due to cell distortion and lysis (*Arnold, Matsche & Rosemary, 2014*). Manual cell counts are necessitated by the morphology of fish cells, which is time and experience-intensive, and is not feasible in large-scale field studies given the number of samples collected. However, the method of formalin fixation previously developed has been proven effective in the hemolymph of lobster (*Basti et al., 2010*), and the heparinized whole blood of select elasmobranchs, and striped bass (*Arnold, Matsche & Rosemary, 2014*). The current study demonstrates the successful use of 10% neutral buffered formalin in another teleost species, the largemouth

bass. Additionally, sample stability was maintained for up to 3.5 months in this study, with no evidence of adverse effects. The extension of cellular morphologic stability provides potential for samples to be transported to diagnostic laboratories for evaluation by trained and experienced analysts for consistency. This technique is revolutionary and will promote the acquisition of precise hematologic information in various fish species that may otherwise remain absent in the veterinary literature.

There are a number of limitations inherent to this study including the presumption that the externally healthy fish were truly disease free, comparison of presumably isolated genetic lineages, the sample population age and sex was undefined, and the weight and length was an unknown variable prior to sample collection. Additionally, the overall small population size was particularly limited by the confines of the single-day angling tournament, even with the new formalin fixation technique for sample preservation. Nevertheless, this prompts a future direction of study towards more extensive hematologic comparison with use of the formalin fixation technique to enhance sample size, while conserving result precision. Seasonality has been documented to impact hematocrit and hemoglobin levels of striped bass from the same reservoir, whereby those sampled in the spring had significantly higher values compared to those sampled in the summer (*Tisa, Strange & Peterson, 1983*). This finding opposes the data presented here, such that the captive-raised fish were sampled in the spring, yet had lower hematocrit values compared to their wild counterparts sampled in the fall. For this reason, it is less likely that season is the underlying reason for the significant differences observed in packed cell volume and total solids. However, seasonal impact on the leukocyte count is unknown and should not be overlooked.

## CONCLUSIONS

This study concludes that wild LMB captured via tournament angling have higher packed cell volume and total solids, while demonstrating lower total leukocyte counts when compared to captive-reared individuals. To the authors' knowledge, this is the first report demonstrating the use of heparinized whole blood preservation with 10% neutral buffered formalin for hematologic evaluation of LMB via manual hemocytometer. Further investigation into the influence of season, and exploration of an increased population size is considered essential for expanding our understanding of LMB hematology. Implementation of the formalin fixation technique to extend the stability of whole blood samples should be regarded as a critical component of fish hematologic research moving forward.

## ACKNOWLEDGEMENTS

Numerous NC State-CVM students, staff, and faculty contributed to the collection of the data presented in this report. The authors also thank Bass Pro Shops and Fishers of Men National Tournament Trail for their support in provision of the tournament-caught fish, Foster Lake and Pond Management, Inc. for their support in provision of the captive-reared fish, and the NC State-CVM Clinical Pathology laboratory for their assistance in sample processing techniques and expertise. We thank Cory Sims for assistance in microphotograph capture and Jill Arnold for her expertise and input.

### Funding

The authors received no funding for this work.

### Competing Interests

The authors declare there are no competing interests.

### Author Contributions

- Michelle C. Whitehead conceived and designed the experiments, performed the experiments, analyzed the data, prepared figures and/or tables, authored or reviewed drafts of the paper, approved the final draft.
- Chelsey L. Vanetten conceived and designed the experiments, performed the experiments, approved the final draft, laboratory analysis (manual differential and hemocytometer analysis).
- Yaxin Zheng analyzed the data, approved the final draft.
- Gregory A. Lewbart conceived and designed the experiments, performed the experiments, authored or reviewed drafts of the paper, approved the final draft.

### Animal Ethics

The following information was supplied relating to ethical approvals (i.e., approving body and any reference numbers):

The study protocol was approved by the Institutional Animal Care and Use Committee at North Carolina State University College of Veterinary Medicine (Protocol #18-009-O).

### Data Availability

The raw measurements are available in Files S1 and S2 for wild-caught and captive-raised largemouth bass, respectively.

### Supplemental Information

Supplemental information for this article can be found online at http://dx.doi.org/10.7717/peerj.6669#supplemental-information.

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
