# Peer review of "Hematological parameters in largemouth bass (Micropterus salmoides) with formalin-preservation: comparison between wild tournament-caught and captive-raised fish"

_PeerJ, doi:10.7717/peerj.6669_

## Round 0.1 · original submission · Minor Revisions

Reviewer 1 has provided very detailed feedback on all aspects of the manuscript, which need to be addressed. Reviewer 2 has also expressed concern regarding the potential effect of cortisol, which is not being measured in this study.

In addition, the authors should discuss more clearly the limitations of comparing differences between captive and wild-caught fish resultant of lineage/environment rather than as a result of the catecholamine/cortisol stress responsiveness following angling (RBC volumetric changes; e.g., MCV, MCHC, Hct, etc.), and the altered density of fish and increased pathogen exposure in captive conditions (elevated leukocyte counts).

·

Basic reporting

In general, the author’s use of language is clear and unambiguous. There are a couple of minor awkward phrases, which will be identified below. The abstract is concise and follows accepted format. Enough information is presented so that the reader can grasp the intent, methods and results. However, I would suggest that more emphasis is placed in the abstract and elsewhere on the relatively new hematologic procedures used and the benefits that the procedure provides for field studies such as this one. While the authors rightly point out that there is a relative paucity of data on LMB hematology, I think this study could highlight the benefits of the 10% NBF treatment method to extend specimen viability in studies where it would be difficult to perform cell counts in a timely manner. The slightly greater value of this article might be the demonstration of the specimen preservation techniques, which encourage more research in this area. Perhaps introducing the need for these techniques in the background section and/or discussion of the abstract would enhance the manuscript.

The sections of the manuscript including introduction, materials and methods, results and discussion is well constructed following scientific formatting, concise, and well written. I generally found the tables and plate of microphotographs well-structured and easily interpreted. The authors provide a thorough introduction that provides relevant background information and problems associated with hematology of fish and other poikilotherms. However, the authors did not present their hypotheses correctly (see lines beginning on 110). They state that their null hypothesis is that there would be a disparity between treatment groups. A null hypothesis in experimental design is that there is NO difference between or among treatment groups. Predicting a significant difference in leukocyte counts would be an alternative hypothesis, of which there may be several. The authors do need to rethink and restate their hypotheses.

The authors generally provide enough information in the materials and methods, including appropriate references so that the procedures they used are understood. I would have liked to see a bit more details on how they handled individual blood specimens during the fishing tournament. How soon after blood smears and PCV preparations did the authors mix and dilute whole blood into formalin? They state that pcv and total solids were performed within 8 hours, but it is more important to know how the authors handled remaining whole blood (chilled? agitated?) and the total time until blood-formalin dilutions were prepared. The authors need to provide some additional information on this topic.

The results are presented concisely and clearly. However, a little rewriting for the benefit of the reader may be in order. I would suggest rather than state that results are “presented in tables” that the authors provide a short summary of relevant results in the text; such as “the median weights and lengths of fish in each treatment group were not significantly different (Table 1)”, or, “the median weights and lengths of wild fish were greater than captive fish, but the differences were not significant (Table 1).” The authors should state the most important results and let the readers then evaluate the tables, images, etc. This is more compelling and interesting reading than simply providing a list of tables.

The discussion raises interesting points and presents the data of this study in appropriate context with previous work. However, the authors used a relatively new technique to extend specimen viability that many fish health workers may not know about. There needs to be a paragraph of discussion on the use of this technique including benefits/limitations and the author’s experiences thus far. Do the authors have any comments on the usefulness or practicality of this technique during a field study in which it may not feasible to perform hemocytometer counts in a timely manner? To me, increasing awareness of the 10%NBF technique to a broader community that may find it useful in fish health surveillance or research may be just as important, or more, then the actual results. I’d like to read what the authors think since they are well-versed in hematological practices.

I have further comments regarding the discussion, which will appear below (general comments section).

Experimental design

I find the experimental design to be sound in general. The results of this study provide useful information and comparison between captive fish and those that undergo the stresses of angling in a tournament setting. The experimental design is limited in that a direct causal relationship of angling on hematologic parameters cannot be determined, and the authors were correct in not overstating conclusions.

Validity of the findings

The conclusions reached by the authors based on the methods employed and results obtained are valid and will provide useful information on LMB responses to the stresses of angling.

Additional comments

25 Background: why use 10%NBF?
42 I don’t necessarily agree that the results indicate superior health in wild fish. It may be, but here are some other possibilities. The decreased leukocyte count in wild fish may be a leukopoenia, which is a common stress reaction in animals. Also, you detected a significantly elevated PCV in wild fish but not in RBC. To me this suggests a shift in plasmatic water resulting in hypovolemia, or in other words, a “hemo-concentration”. The elevated PCV in wild fish does not necessarily mean increased oxygen carrying capacity as the RBC numbers were comparable. There may be the same number of RBC but in a smaller volume of plasma in wild fish. Shifts in water to/from plasma as a result of osmotic changes to the gills and other tissues are a common stress reaction and should not be overlooked. However, maintaining fish in aquaculture is also no guarantee of superior health. There are a lot of stresses encountered in captivity based on type of containment, stocking density, DO levels, etc. There are papers that demonstrate wide differences in hematology and plasma chemistry based on differences in conditions. The following paper is 1 example. There are others.

Matsche MA, Arnold J, Jenkins E, Townsend H, Rosemary K (2014) Determination of hematology and plasma chemistry reference intervals for 3 populations of captive Atlantic sturgeon (Acipenser oxyrinchus oxyrinchus). Vet Clin Pathol 43:387-396

So while this study does provide useful information, it is important to not overstate conclusions. The authors have stated more than once that numerous factors influence hematology and there are likely marked differences in conditions and water quality between wild and captive fish. The authors need to rethink and restate this discussion.

44 “…increased sample size…” This is a primary benefit of specimen preservation!

110 Your null hypothesis should be N0: wild fish parameter = captive fish parameter.
Alternatives may include: N1: wild fish parameter > captive fish parameter, and
N2: wild fish parameter < captive fish parameter.

140 No need to make assumptions. State that the fish sampled in this study appeared robust with no external lesions, if that was the case. Save any discussion of cryptic health problems for the discussion.

150 Sample preparation in formalin. I need to see more details as to how long before formalin dilution was performed and how specimens were handled prior to dilution.

174 Restructure awkward sentence. Suggest starting paragraph with “Hotelling’s t-test was performed to compare…”, or similar.

191 Delete the word “contrarily”.
Discussion in general As mentioned before, there needs to be discussion of the benefits/limitations of the 10%NBF technique to preserve specimens.

197 and on The authors should restate this section based on changes that are needed regarding the statement of the null and alternate hypotheses.

Tables 1 and 2 Min-Max are presented in parentheses, which distinguishes this information from medians. I would therefore remove parentheses from around samples sizes to avoid confusion. There appears to be enough room to include sample size “n” as a separate column in the tables, but current layout is ok.

Reviewer 2 ·

Basic reporting

The work of Michelle C Whitehead et al. studies the Evaluation of A Hematological parameters in largemouth bass (Micropterus salmoides) with formalin-preservation:comparison between wild tournament-caught and captive-raised fish.

The study also included the analysis of the some Hematological indices.

Overall, the study contains good and valuable content; however, the novelty and the data of this article can be presented as a short communication finding, not as a complete article.

But there are several important issues that should be addressed before publication. The coherence of the text must be improved

Experimental design

See below

Validity of the findings

See below

Additional comments

Abstract:

It is suitable

Introduction:

- Line 86-87: “Talking about dietary lipid is excessive and useless.

Materials and Methods:

The weighted average of the two groups is very different, and we can not generalize the results with respect to this weight difference for all categories.

Results:

It is suitable

Discussion:

In line 217: The changes in the count of white blood cells were related to age, size, and weight, but as I explained in the material and method the difference in weight and size in the tested fish is high and no countable results can be obtained.

Lin235-244: Because cortisol has not been measured in the present study, discussion of this is not feasible.

---

## Round 0.2 · accepted · Accept

I consider that you have adequately addressed the reviewers' comments. The additional discussion and alternative interpretations have strengthened the manuscript.

#